# Synergistic Anti-Inflammatory Activity of Lipid-Free Apolipoprotein (apo) A-I and CIGB-258 in Acute-Phase Zebrafish via Stabilization of the apoA-I Structure to Enhance Anti-Glycation and Antioxidant Activities

**DOI:** 10.3390/ijms25105560

**Published:** 2024-05-20

**Authors:** Kyung-Hyun Cho, Ashutosh Bahuguna, Yunki Lee, Sang Hyuk Lee, Maria del Carmen Dominguez-Horta, Gillian Martinez-Donato

**Affiliations:** 1Raydel Research Institute, Medical Innovation Complex, Daegu 41061, Republic of Korea; 2Center for Genetic Engineering and Biotechnology, Ave 31, e/158 y 190, Playa, La Havana 10600, Cuba

**Keywords:** apolipoprotein A-I (apoA-I), high-density lipoproteins (HDLs), CIGB-258 (Jusvinza^®^), carboxymethyllysine (CML), zebrafish, interleukin-6

## Abstract

CIGB-258, a 3 kDa peptide from heat shock protein 60, exhibits synergistic anti-inflammatory activity with apolipoprotein A-I (apoA-I) in reconstituted high-density lipoproteins (rHDLs) via stabilization of the rHDL structure. This study explored the interactions between CIGB-258 and apoA-I in the lipid-free state to assess their synergistic effects in the structural and functional enhancement of apoA-I and HDL. A co-treatment of lipid-free apoA-I and CIGB-258 inhibited the cupric ion-mediated oxidation of low-density lipoprotein (LDL) and a lowering of oxidized species in the dose-responsive manner of CIGB-258. The co-presence of CIGB-258 caused a blue shift in the wavelength of maximum fluorescence (WMF) of apoA-I with protection from proteolytic degradation. The addition of apoA-I:CIGB-258, with a molar ratio of 1:0.1, 1:0.5, and 1:1, to HDL_2_ and HDL_3_ remarkably enhanced the antioxidant ability against LDL oxidation up to two-fold higher than HDL alone. HDL-associated paraoxonase activities were elevated up to 28% by the co-addition of apoA-I and CIGB-258, which is linked to the suppression of Cu^2+^-mediated HDL oxidation with the slowest electromobility. Isothermal denaturation by a urea treatment showed that the co-presence of CIGB-258 attenuated the exposure of intrinsic tryptophan (Trp) and increased the mid-points of denaturation from 2.33 M for apoA-I alone to 2.57 M for an apoA-I:CIGB-258 mixture with a molar ratio of 1:0.5. The addition of CIGB-258 to apoA-I protected the carboxymethyllysine (CML)-facilitated glycation of apoA-I with the prevention of Trp exposure. A co-treatment of apoA-I and CIGB-258 synergistically safeguarded zebrafish embryos from acute death by CML-toxicity, suppressing oxidative stress and apoptosis. In adult zebrafish, the co-treatment of apoA-I+CIGB-258 exerted the highest anti-inflammatory activity with a higher recovery of swimming ability and survivability than apoA-I alone or CIGB-258 alone. A co-injection of apoA-I and CIGB-258 led to the lowest infiltration of neutrophils and interleukin (IL)-6 generation in hepatic tissue, with the lowest serum triglyceride, aspartate transaminase, and alanine transaminase levels in plasma. In conclusion, the co-presence of CIGB-258 ameliorated the beneficial functionalities of apoA-I, such as antioxidant and anti-glycation activities, by enhancing the structural stabilization and protection of apoA-I. The combination of apoA-I and CIGB-258 synergistically enforced the anti-inflammatory effect against CML toxicity in embryos and adult zebrafish.

## 1. Introduction

High-density lipoproteins (HDLs) have antioxidant and anti-inflammatory activity to suppress the oxidation of low-density lipoproteins (LDLs) and the production of pro-inflammatory cytokines in the acute-phase response in the blood and lungs [1,2]. Apolipoprotein A-I (apoA-I), a major protein component in HDL of around 70%, is mainly responsible for the antioxidant, anti-inflammatory, and anti-tumorigenic activities with cholesterol efflux ability [3,4]. ApoA-I can reduce systemic and lung inflammation in the acute phase by modulating innate and adaptive immunity [5]. In the hyperinflammatory state, such as critical phase of COVID-19 with the acute elevation of tumor necrosis factor (TNF)-α and interleukin (IL)-6, serum concentrations of HDL-C and apoA-I significantly decreased and were negatively correlated with COVID-19 severity [6,7]. Furthermore, the decreased apoA-I concentration was positively associated with the severity of COVID-19. It was negatively correlated with the production of IL-6 and high-sensitive C-reactive protein (CRP) [7], suggesting that apoA-I itself could exert anti-inflammatory and antiviral activities [8].

On the other hand, HDL can become dysfunctional with a decrease in cholesterol content and an increase in oxidation and glycation, particularly in the acute-phase response [9,10]. In particular, the loss of HDL functionality is directly associated with the alteration of apoA-I by glycation, nitration, and myeloperoxidase-mediated oxidation [11]. The higher modification extent of apoA-I was directly correlated with the larger loss of cholesterol efflux ability, which is linked with damage to HDL particle formation. Non-enzymatic glycation impaired the structural stability and functionality of apoA-I, such as decreased HDL particle size, cholesterol efflux ability, and paraoxonase activity [12,13]. Interestingly, the apoA-I half-life was longer with lower glycated hemoglobin (HbA_1c_) levels. Specifically, the half-life of glycated apoA-I was three times shorter than that of native apoA-I [13]. Among the advanced glycation end products (AGE), carboxymethyllysine (CML), a glycoxidation product of glycated lysine residues, caused an increase in atrial stiffness and pulmonary fibrosis [14,15]. In the same context, previous reports showed that a CML treatment led to significant increase in the extent of glycation of HDL and apoA-I, accompanied by severe proteolytic degradation [16,17]. High levels of CML in serum were also identified in patients with type 2 diabetes mellitus (T2DM) and cardiovascular diseases, who had extremely high inflammatory cytokines [18,19].

These studies helped develop a new pharmaceutical agent to treat the cytokine storm and hyperinflammation in acute and chronic inflammatory disease by protecting HDL and apoA-I from glycoxidation attack, maximizing its anti-infection and anti-inflammation activity. CIGB-258 (Jusvinza^®^), an altered peptide ligand composed of 27 amino acids with a molecular weight of 2987, is derived from heat shock protein HSP60. It has demonstrated protective effects on HDL and apoA-I against proteolytic degradation caused by glycation and oxidation, specifically, CIGB-258 dose-dependent protection for HDL and apoA-I from CML-induced glycation by stabilizing their protein structure and enhancing their antioxidant capacity. Moreover, CIGB-258 could bind strongly with apoA-I and transthyretin (TTR) in human serum from affinity chromatography [20], but the precise mechanism and purpose are unclear.

A previous study reported that CIGB-258 can effectively bind phospholipids and cholesterol, stabilizing apoA-I within the reconstituted HDL (rHDL) structure and leading to the formation of larger rHDL particles. The rHDL containing CIGB-258 exhibited improved in vitro antioxidant ability against LDL oxidation, enhanced anti-glycation activity to protect HDL, and showed in vivo anti-inflammatory effects against CML toxicity in embryos and adult zebrafish. The incorporation of apoA-I and CIGB-258 in the lipid-bound state (rHDL) resulted in synergistic interactions that enhanced the structural integrity and functional performance of rHDL in a dose-dependent manner. On the other hand, there are no reports of an interaction between apoA-I and CIGB-258 in the lipid-free state, either synergistically or in an uncooperative manner, depending on the increase in CIGB-258 in the mixture. The study aims to compare the putative collaboration effect between the apoA-I and CIGB-258 at molar ratios of 1:0, 1:0.1, 1:0.5, and 1:1 to exert in vitro antioxidant activity and anti-glycation activity by stabilizing the lipid-free apoA-I structure. The apoA-I and CIGB-258 mixture were administrated to zebrafish embryos and adults to evaluate in vivo anti-inflammatory activities in the presence of CML, which can cause acute embryotoxicity, developmental defects, and acute paralysis and inflammatory death.

## 2. Results

### 2.1. Antioxidant Activity of Lipid-Free apoA-I and CIGB-258

Native LDL (lane N) exhibited a distinct band intensity and the slowest electromobility (as indicated by the black arrow), while the band intensity of oxidized LDL (oxLDL, lane O) in the presence of Cu^2+^ (final 1 μM) almost disappeared with the fastest electromobility (as indicated by the red arrow) (Figure 1A). Interestingly, the co-treatment of apoA-I and CIGB-258 inhibited the LDL oxidation with a thicker band intensity than oxLDL (lanes 1–4), suggesting that LDL was protected from the cupric ion-mediated oxidation. Quantification of the oxidized species in each LDL showed that native LDL had the lowest level of malondialdehyde (MDA), while oxLDL had the highest level of MDA, 11-fold higher MDA than native LDL (Figure 1B). Interestingly, a co-treatment with apoA-I and CIGB-258 at 1:0, 1:0.1, 1:0.5, and 1:1 molar ratio lowered the MDA level by 27% (*p* < 0.05), 44% (*p* < 0.01), 57% (*p* < 0.001), and 71% (*p* < 0.001) lower than that of oxLDL, respectively. These results suggest that a co-treatment of apoA-I and CIGB-258 up to a 1:1 molar ratio inhibited oxidation to protect the LDL band and lower the oxidized species via the putative stabilization of the apoA-I structure.

### 2.2. Enforcement of Antioxidant Ability of HDL by Co-Presence of CIGB-258

Native LDL (lane N) showed the highest distinct band intensity with the slowest electromobility (Figure 2A), while the band intensity of oxidized LDL (lane O) nearly vanished alongside the aggregated band at the loading position, as highlighted by the red arrowhead. The HDL_2_ alone (lane 1) and HDL_3_ alone (lane 5) treatments inhibited the oxidation and degradation of LDL, suggesting that HDL possesses adequate inhibition activity against LDL oxidation. Conversely, the co-presence of CIGB-258 led to stronger antioxidant ability of HDL_2_ (lanes 2–4) and HDL_3_ (lanes 6–8), in a dose-dependent manner, of CIGB-258. The higher CIGB-258 dosage caused the slower electromobility of LDL with more distinct band intensity under the same amount of HDL, suggesting that the co-presence of CIGB-258 can elicit the antioxidant ability of HDL.

Oxidized LDL showed a 20-fold higher MDA level than native LDL (Figure 2B), while HDL_2_ or HDL_3_ alone-treated LDL showed 37% and 30% lower MDA levels. These results suggest that HDL_2_ and HDL_3_ alone could exert adequate inhibition activity against LDL oxidation via HDL-associated paraoxonase activity. On the other hand, the co-presence of CIGB-258 at an apoA-I:CIGB-258 molar ratio of 1:1 reduced the MDA level (up to 57% (in HDL_2_) and 46% (in HDL_3_) lower than oxLDL) in a dose-responsive manner. The findings imply that HDL and CIGB-258 exhibited synergistic antioxidant activity against LDL oxidation to protect LDL particles from proteolytic degradation and aggregation (Figure 2A). A further increase in the CIGB-258 ratio in HDL resulted in the detection of a lower MDA level in LDL (Figure 2B), suggesting that CIGB-258 could promote the antioxidant ability of HDL.

### 2.3. HDL-Associated Paraoxonase Activity with CIGB-258

As depicted in Figure 3A, during 90 min incubation, the native HDL_3_ showed 2.5-times higher paraoxonase (PON) activity than HDL_2_, suggesting that HDL_3_ is the principal source of PON activity in total HDL. The addition of an apoA-I:CIGB-258 mixture to HDL_2_ and HDL_3_ elevated the PON activity as the CIGB-258 content was increased. The addition of a 1:1 mixture (apoA-I:CIGB-258) into HDL_2_ and HDL_3_ resulted in the highest PON activity, approximately 13% and 28% higher than that of native HDL_2_ or HDL_3_ alone, respectively. In contrast, the 1:0.5 mixture (apoA-I:CIGB-258) resulted in the second-highest PON activity of HDL_2_ and HDL_3_, approximately 14% and 18% higher than HDL alone. Interestingly, the addition of apoA-I alone or CIGB-258 alone showed a similar increase in PON activity, approximately 5–7% and 3–5% higher than HDL_2_ and HDL_3_ alone. These results suggest that a combination of apoA-I and CIGB-258 had the highest PON activity in HDL_2_ and HDL_3_, with synergistic activity dependent on the CIGB-258 content.

In the same context, native HDL_2_ displayed the restarted electromobility, characterized by a distinct sharp band intensity (Figure 3B), highlighted by the black arrow (lane N). Conversely, HDL_2_ subjected to cupric ion treatment exhibited rapid electromobility, migrating to the gel bottom with the diminished and diffused band intensity (lane O), as denoted by the red arrow, attributed to proteolytic degradation, and improved negative charge. On the other hand, adding the apoA-I and CIGB-258 mixture induced slower electromobility with a stronger band intensity, depending on the increase in CIGB-258 content (lanes 1–4). In particular, the co-addition of a 1:1 mixture (apoA-I:CIGB-258) into HDL_2_ resulted in the slowest electromobility among the oxidized lipoproteins (lane 4), suggesting that a greater elevation in CIGB-258 content induced more resistance to HDL_2_ oxidation.

### 2.4. Stabilization of apoA-I Structure by Co-Presence of CIGB-258

In the absence of urea, lipid-free apoA-I alone displayed 336.5 nm of WMF, while the addition of CIGB-258 caused a blue shift of apoA-I toward 335.9, 334.9, and 334.7 nm for 1:0.1, 1:0.5, and 1:1, respectively, as depicted in Figure 4A. These findings imply that the co-presence of CIGB-258 prompted the migration of intrinsic Trp, particularly Trp 108, within apoA-I towards the hydrophobic environment, possibly through a presumed interaction between the amphipathic helix regions of apoA-I and CIGB-258.

Interestingly, SDS-PAGE displayed the fact that the intensity of the apoA-I band was increased by the co-presence of CIGB-258 in a dose-responsive manner up to a 1:1 molar ratio (Figure 4B). Compared to a 1:0 molar ratio of apoA-I:CIGB-258, the 1:0.5 and 1:1 apoA-I:CIGB-258 mixtures showed an up to 2.1- and 2.2-fold band intensity, respectively, due to the addition of CIGB-258. In lanes 3 and 4, the CIGB-258 band was detected at the base of the gel, as highlighted by the red arrowhead. In particular, a 1:1 blend of apoA-I and CIGB-258 showed the strongest band intensity (lane 4, Figure 4B). The results imply that the co-presence of CIGB-258 induced more stabilization of apoA-I structure to protect against the denaturation and proteolysis of apoA-I.

The isothermal denaturation of lipid-free apoA-I alone (1:0) by a urea treatment caused a 20.4 nm increase in WMF from 336.5 nm (without urea) to 356.9 nm (7 M urea), suggesting exposure of intrinsic Trp toward the hydrophilic phase by unfolding of the secondary structure. The sigmoidal curve of WMF showed a typical α-helix-enriched protein of apoA-I. Until 2 M urea treatment, 9 nm of WMF was increased from the baseline (0 M urea), indicating apoA-I was resistant to denaturation at 1–2 M urea (Figure 4A) with q mid-point of denaturation (D_1/2_) of 2.33 M of urea (Table 1). Conversely, the addition of CIGB-258 into apoA-I displayed a slower enhancement in the WMF upon the same urea exposure, a 6–7 nm rise in WMF with the 2 M urea exposure, indicating less exposure of intrinsic Trp via more resistance to denaturation. At 3 M urea treatment, the WMF of apoA-I was red-shifted to 351.9 nm, 351.4 nm, 347.3 nm, and 347.8 nm for apoA-I:CIGB-258 molar ratios of 1:0, 1:0.1, 1:0.5, and 1:1, respectively. These results suggest that the co-presence of CIGB-258 confers resistance of apoA-I against chaotropic agent-induced denaturation by stabilizing the α-helical domains and tertiary structure.

Regression analysis showed that the mid-points of denaturation were increased up to 2.35 M, 2.57 M, and 2.56 M of urea for apoA-I:CIGB-258 ratios of 1:0.1, 1:0.5, and 1:1, respectively, suggesting less exposure of intrinsic Trp via more stabilization of the α-helix domains of apoA-I. These results indicate that the secondary and tertiary configuration of apoA-I could be improved by the co-presence of CIGB-258 via the putative helix–helix interactions. The stabilization of apoA-I (Figure 4A) was linked with more protection of the apoA-I band intensity, with a concomitant increase in CIGB-258 (Figure 4B).

### 2.5. Anti-Glycation Activity of CIGB-258 against CML-Induced apoA-I Glycation

The CML exposure of lipid-free apoA-I caused the largest increase in yellowish fluorescence intensity (FI), 8.1-fold higher FI than apoA-I alone during 72 h, as shown in Figure 5A. Conversely, the co-treatment of CIGB-258 caused a smaller increase in FI in a dose-dependent manner: 3%, 17%, and 25% reduction for apoA-I:CIGB-258 molar ratios of 1:0.1, 1:0.5, and 1:1, respectively. With the largest increase in glycation extent by the CML treatment (final 200 mM), the WMF of the glycated apoA-I (349.5 nm) showed a 5.5 nm red shift at 72 h incubation compared to the baseline WMF at 0 h (344.0 nm), as shown in Figure 5B. Through glycation, the intrinsic Trp of apoA-I was more exposed to the hydrophilic phase, due to unfolding of the α-helix domain and an unstable tertiary structure formed. Nevertheless, the co-presence of CIGB-258 attenuated the increase in WMF, in a dose-dependent manner, after 72 h incubation: 346.9 nm, 345.8 nm, and 343.5 nm for apoA-I:CIGB-258 molar ratios of 1:0.1, 1:0.5, and 1:1, respectively. The results imply that the co-presence of CIGB-258 inhibited CML-mediated glycation to stabilize the α-helix domain and tertiary structure with the maintenance of Trp toward the hydrophobic phase.

### 2.6. Zebrafish Embryo Protection against CML-Toxicity

A microinjection of CML (500 ng) into zebrafish embryos led to acute death and the lowest survivability (21 ± 2% survivability) at 24 h post-injection (Figure 6). In contrast, the PBS-injected embryo showed the utmost survivability of 89 ± 3% (Figure 6). In the presence of CML, however, a co-injection of apoA-I (1.4 ng) alone or CIGB-258 alone (143 pg) resulted in a higher survivability of around 37 ± 5% and 33 ± 1%, respectively, suggesting that either apoA-I or CIGB-258 possessed adequate anti-inflammatory activity to neutralize the CML toxicity. Furthermore, in the presence of CML, co-injection of an apoA-I:CIGB-258 mixture yielded significantly higher embryo survivability, in a dose-dependent manner, of CIGB-258: 47 ± 3% (*p* < 0.01), 78 ± 2% (*p* < 0.001), and 82 ± 2% (*p* < 0.001) for apoA-I:CIGB-258 molar ratios of 1:0.1, 1:0.5, and 1:1, respectively. These findings indicate that the presence of CIGB-258 significantly improved the anti-inflammatory capabilities of apoA-I, effectively reducing acute embryo mortality through a potential synergistic effect of structural stabilization.

The stereo image of the embryos showed that the PBS-alone group revealed a normal developmental speed and morphology at 5 h, 24 h, and 48 h (Figure 7A, photograph a). All the embryos in the PBS-alone group displayed the primordium-6 stage with the darkest eye pigmentation and tail elongation, along with the highest hatching (~78%) and somite counts (~34.6 ± 0.3) at 48 h post-treatment (Figure 7A,B). Conversely, the CML+PBS-injected embryo showed the most severe embryonic defects, with the least embryo hatching (~4%) (photograph b). The co-injection of apoA-I alone improved the CML-altered survivability (photograph c) and substantially improved the hatching (~18%) and somite counts (~22 ± 0.3) (Figure 7A,B). Likewise, similar results were observed for the only CGB-258-alone (0:1) injected group (photograph g). Interestingly, the 1:0.1 and 1:0.5 mixture of the apoA-I and CIGB-258 groups showed a much faster developmental speed (photograph d and e) than the apoA-I (1:0) and CIGB-258-alone (0:1) groups, with substantial high hatching and somite counts (Figure 7A,B). On the other hand, co-injection of a 1:1 mixture of (apoA-I and CIGB-258) resulted in the most improved developmental speed and morphology (photograph f) altered by CML, and all embryos showed the primordium-6 stage with the darkest eye pigmentation and tail elongation. As compared to the CML-injected group, the 1:1 mixture of (apoA-I and CIGB-258) resulted in 18-fold higher embryo hatching (~61%) and 32-fold higher somite counts (~32 ± 0.3) (Figure 7A,B). These results suggest that the co-presence of CIGB-258 helped to protect the embryos from CML-mediated embryotoxicity, in a dose-dependent manner, of CIGB-258. 

AO staining to detect cellular apoptosis showed that the CML-alone group had a 5.3-fold larger increase in apoptosis than the PBS group, suggesting that the CML injection induced acute cell death (Figure 7B). On the other hand, the co-injection of a 1:1 mixture resulted in the least apoptosis (~79% less apoptosis than the CML-alone group), while the apoA-I-alone (1:0) group showed no significant reduction: ~15% lower than the CML+PBS group. Interestingly, the (1:0.1)- and (1:0.5)-mixture groups showed a more significant decrease in apoptosis (~23% and 73% reduction, respectively) than the CML+PBS group. Hence, the cytoprotective effect of apoA-I was enhanced by the co-presence of CIGB-258 in a dose-dependent manner.

DHE staining to detect ROS showed that the CML+PBS injection caused a 3.3-fold increase in ROS production compared to PBS alone (Figure 7B). A co-injection of apoA-I alone (1:0) resulted in the weakest activity for reducing ROS generation (~12% lower than the CML+PBS group). A co-injection of the 1:1 mixture had the strongest activity in lowering ROS generation (~66% reduction of ROS). The co-injection of 1:0.1 or 1:0.5 mixtures also reduced ROS production (~31% and ~63% lower than the CML+PBS group, respectively). Overall, all mixtures induced adequate protective activity against the CML toxicity, in a dose-dependent manner, of CIGB-258. The 1:1 mixture exerted the most potent activity in recovering the highest survivability and fastest development.

### 2.7. Anti-Inflammatory Effect against CML-Induced Acute Toxicity

An intraperitoneal (IP) injection of 250 μg of CML into adult zebrafish (~16 weeks old) (to a final concentration of approximately 3 mM in zebrafish body weight of around 300 mg) caused acute paralysis at 30 min post-injection (Figure 8A). On the other hand, in the presence of CML, a co-injection of either apoA-I (8.5 μg, final 1 μM) or CIGB-258 (0.9 μg, final 1 μM) resulted in higher swimming ability: ~10–13% and ~45–47% at 30 min and 60 min post-injection, respectively. Furthermore, the apoA-I:CIGB-258 (1:1) group showed the highest swimming ability around 25 ± 3% and 70 ± 5% at 30 min and 60 min post-injection, respectively (Figure 8A), suggesting that a combination of apoA-I and CIGB-258 exhibited synergistic anti-inflammatory activity to suppress paralysis and the acute toxicity posed by CML.

The CML+PBS group showed the lowest survivability (~58% and ~47% at 1 h and 3 h post-injection, respectively), indicating severe lethal toxicity of CML (Figure 8B). On the other hand, the CML+apoA-I and CML+CIGB-258 groups showed comparable survivability (~70–80% and 65–70% at 1 h and 3 h post-injection, respectively), suggesting that apoA-I and CIGB-258 had adequate anti-inflammatory activity, to a similar extent. On the other hand, the CML+apoA-I+CIGB-258 group showed the highest survivability, ~92% and ~85% at 1 h and 3 h post-injection (Figure 8B), indicating synergistic anti-inflammatory activity to maximize the survivability between apoA-I and CIGB-258.

Although the PBS-alone group (photograph a) exhibited an active swimming pattern (Appendix A), all zebrafish in the CML+PBS group (photograph b) could not swim and were lying down on the bottom of the tank with occasional quivering (Figure 8C) despite being still alive at 30 min post-injection. At 1 h post-injection, 25% of the fish could swim again, with 58% survivability in the CML+PBS group, but the swimming pattern involved wobbling, seizure, and uncontrollable vertical movements (Appendix A). In contrast, the CML+apoA-I group showed an enhanced recovery of swimming ability of ~45 ± 3% and survivability of ~70% at 1 h post-injection, displaying a more improved swimming pattern, albeit with wobbling and seizure still detected (Appendix A). Similarly, the CML+CIGB-258 group showed an improved recovery of their swimming ability (~46 ± 10%) at 1 h post-injection with 80 ± 4% survivability (Appendix A). On the other hand, the CML+apoA-I+CIGB-258 group showed the fastest recovery of swimming ability of ~70 ± 6% and highest survivability of ~92 ± 3% at 1 h post-injection with the most active and natural swimming pattern (Appendix A). At 3 h post-injection, the CML+apoA-I+CIGB-258 group showed the highest survivability of ~85 ± 3%, while the CML+apoA-I and CML+CIGB-258 groups showed 65 ± 6% and 70 ± 5% survivability, respectively. These results suggest that a combination of apoA-I and CIGB-258 had a synergistic effect on improving swimming ability and survivability against the acute toxicity of CML.

### 2.8. Histologic Examination of Hepatic Tissue

The H&E analysis of the liver section as documented in Figure 9, displayed the hepatological changes in the different groups. A massive neutrophil infiltration was observed in the PBS+CML group, which was, significantly, 15.2-fold (*p* < 0.001) higher than the neutrophil counts observed in the PBS-injected control group (Figure 9B). The CML-induced neutrophil infiltration was effectively countered by the injection of apoA-I, CIGB-258 and apoA-I+CIGB-258, evident in a significant 2.8-fold, 2.7-fold and 8.8-fold reduced neutrophil count in the respective groups, as compared to the CML-injected group. However, the most promising effect was exerted by apoA-I+CIGB-258, as evident in significantly ~3-fold (*p* < 0.05) reduced neutrophil counts in the hepatic section of the apoA-I+CIGB-258- treated group compared to the apoA-I- and CIGB-258-treated groups, testifying enhanced protective activity against CML-induced hepatotoxicity when using the combination of apoA-I+CIGB-258.

### 2.9. Extent of ROS Production and Apoptosis in Hepatic Tissue

AO staining revealed the PBS-alone group to have the weakest green fluorescence (photo a1, Figure 10A), ~8 ± 2%, while the CML+PBS group had the highest green fluorescence area (photo a2), of ~41 ± 2% (Figure 10B), showing the greatest extent of cellular apoptosis, indicating that an IP injection of CML causes acute apoptosis. In contrast, the CML+apoA-I group showed a 37%-lower AO-stained area (*p* < 0.001) than the CML+PBS group, while the CML+CIGB-258 group exhibited a 53%-lower AO-stained area (*p* < 0.001). Interestingly, the CML+apoA-I+CIGB-258 group showed the smallest AO-stained area (~13 ± 1%), which was 67% lower than the CML+PBS group (Figure 10A,B). These results suggest that a co-injection of both apoA-I and CIGB-258 was more effective in preventing cellular apoptosis, even though injection of either apoA-I or CIGB-258 was also effective in preventing apoptosis in the presence of CML.

DHE staining also revealed the PBS-alone group to have the weakest red fluorescence of ~9 ± 1% (photo a2), while the CML+PBS group had the highest red fluorescence intensity of ~40 ± 2% (photo b2), representing the highest ROS production (Figure 10). By contrast, the CML+apoA-I group (photo c2) showed a 37%-lower DHE-stained area (*p* < 0.001) than the CML+PBS group, while the CML+CIGB-258 group (photo d2) exhibited a 47%-lower AO stained area (*p* < 0.001). Interestingly, the CML+apoA-I+CIGB-258 group (photo e2) showed the smallest AO-stained area (~12 ± 1%), which was 70% lower than the CML+PBS group. These results suggest that a co-injection of apoA-I and CIGB-258 was more effective in preventing ROS production, even though an injection of either apoA-I or CIGB-258 was also effective in preventing oxidative stress in the presence of CML. Overall, a combination of apoA-I and CIGB-258 induced remarkable protection of hepatic tissue from cellular apoptosis and ROS production.

### 2.10. Immunohistochemistry for IL-6 Detection in Hepatic Tissue

The immunohistochemical detection of interleukin (IL)-6 in the hepatic tissue revealed the PBS-alone group to have the smallest stained area (photo a1) and red conversion area (photo a2) of ~3.7% (Figure 11A), while the CML+PBS group exhibited the largest stained area (photo b1) and red conversion area (photo b2) of ~22.9% (Figure 11B). On the other hand, the CML+apoA-I group showed a 15.2% IL-6-stained area, which was a ~33% further reduction than the CML+PBS group (Figure 11B), suggesting that co-injection of apoA-I (8.5 μg/zebrafish, final 1 μM) could alleviate the inflammatory response to lower the IL-6 level. Interestingly, the CML+CIGB-258 group showed a more remarkable decrease in IL-6 stained area (photo d1 and d2), a ~7.6% stained area, which was 67% smaller than the CML+PBS group, suggesting that an injection of CIGB-258 (0.9 μg, final 1 μM) was two times more effective in reducing the hepatic IL-6 level of CML-mediated inflammation than apoA-I (final 1 μM). Moreover, the CML+apoA-I+CIGB-258 group showed the smallest IL-6-stained area (photo e1 and e2), ~6.3%, which was 73% lower (*p* < 0.001) than that of the CML+PBS group, indicating the strongest synergistic anti-inflammatory activity through the co-presence of apoA-I and CIGB-258.

### 2.11. Change in the Serum Lipid Profile

After collecting plasma from each zebrafish group, plasma lipid quantification showed that the CML+ PBS group had the highest total cholesterol (TC) and triglyceride (TG) plasma levels, as shown in Figure 12A,B. In particular, the serum TC and TG levels in the CML+PBS group were approximately 1.5-fold and 2.5-fold higher than those of the PBS-alone group, respectively, suggesting an abrupt elevation in TC and TG by the blood infusion of CML. On the other hand, a co-injection of apoA-I or CIGB-258 resulted in a decrease in TC and TG levels to a similar extent: a 37–39% and 60–62% reduction in TC and TG, respectively. These results suggest that apoA-I alone or CIGB-258 alone could lower the plasma TC and TG with the concomitant suppression of the inflammatory cascade. Interestingly, a co-injection of apoA-I and CIGB-258 resulted in the lowest levels of TC and TG: 45% and 65% lower than those of the CML+PBS group, respectively, suggesting that a combination of apoA-I and CIGB-258 had higher lipid-lowering activity.

Quantification of HDL-C in plasma showed that the combined apoA-I and CIGB-258 group showed the highest level of HDL-C (~123 mg/dL), while the apoA-I-alone group and CIGB-258-alone group exhibited a lower level (~69–72 mg/dL). On the other hand, the percentages of HDL-C in TC (HDL-C/TC (%)) were higher in the apoA-I-alone and CIGB-258-alone group, with 27–29% of HDL-C/TC (%), which were higher than that of the CML+PBS group. Surprisingly, a co-injection of the apoA-I and CIGB-258 group showed the highest HDL-C/TC (%), ~54%, suggesting a synergistic effect in increasing the HDL-C (mg/dL) and HDL-C/TC (%).

### 2.12. Change in the Serum AST and ALT

The CML+PBS group showed the highest AST and ALT levels (515 IU/L and 252 IU/L, respectively), which were 4.0-fold and 2.0-fold higher than those of the PBS-alone group, respectively (Figure 13). On the other hand, the CML+apoA-I group showed lower AST and ALT levels (442 IU/L and 197 IU/L, respectively), which were 15% and 22% lower than CML+PBS group, respectively. On the other hand, the CML+CIGB-258 group showed AST and ALT levels of 353 and 175 IU/L, respectively, which were 32% and 31% lower than those of the CML+PBS group, respectively. Interestingly, the CML+apoA-I+CIGB-258 group showed the lowest AST and ALT levels (253 IU/L and 147 IU/L, respectively), which were 51% and 42% lower than those of the CML+PBS group, respectively. Hence, a co-injection of apoA-I and CIGB-258 synergistically ameliorated the hepatic damage, particularly lowering the AST and ALT levels caused by CML toxicity.

## 3. Discussion

CIGB-258 exhibits anti-inflammatory activity against acute toxicity of CML within a normal diet [16] and a high-cholesterol diet (HCD) [17]. In normolipidemic zebrafish, the CIGB-258 group showed higher recovery of swimming ability and survivability than the Infliximab (Remsima^®^) and Tocilizumab (Actemra^®^) groups, with the least hepatic inflammation [16]. In hyperlipidemic zebrafish, a co-injection of CIGB-258 resulted in a 2.2-fold faster recovery of swimming ability than the CML alone with the lowest IL-6 level in hepatic tissue compared to the Infliximab, Etanercept (Enbrel^®^), and Tocilizumab groups [17]. These results suggest that CIGB-258 has similar efficacy to the IL-6 inhibitor rather than the TNF-α inhibitor regarding the lowest IL-6 level, with improvements in survivability and lipid profiles. These results also show good agreement with a previous report that HDL and apoA-I suppress IL-6 production via Toll-like receptor-4 signaling [21,22]. In addition, in the presence of CML in zebrafish embryos, the co-addition of lipid-free apoA-I and CIGB-258 resulted in the highest hatching ratio and somite numbers with increasing CIGB-258 content (Figure 7B).

In addition to HDL, beyond cholesterol trafficking in blood, lipid-free apoA-I is involved in the multi-functional innate immune response and regulation of antiviral activity [23] and anti-inflammatory activity, and has a tumor-suppressive role [14,24]. ApoA-I exerts antiviral activity to inhibit herpes simplex virus-induced cell fusion and prevent viral penetration [23]. Furthermore, apoA-I also displays potent bactericidal activity [25], facilitation of complement-mediating bacterial killing, and protection against the invasion of protozoal parasites, such as trypanosome brucei [26]. The maintenance of higher antioxidant ability and apoA-I content in HDL was reported to be critical for maximizing and preserving the broad spectrum of anti-infection activity [27].

In the current study, CIGB-258 helped apoA-I and HDL induce more antioxidant ability against LDL oxidation in a dose-dependent manner (Figure 1, Figure 2 and Figure 3). During isothermal denaturation by adding urea, the co-presence of CIGB-258 helped stabilize the tertiary structure of apoA-I against proteolytic degradation and Trp exposure (Figure 4 and Table 1). CIGB-258 inhibited the CML-mediated glycation of apoA-I to enhance the structural stabilization via the blue shift in intrinsic Trp (Figure 5). In the presence of CML to induce the acute death of zebrafish embryos, a co-injection of an apoA-I:CIGB-258 mixture helped induce higher survivability and faster developmental speed with lower apoptosis and ROS production according to the CIGB-258 content (Figure 6 and Figure 7). Regarding the acute inflammatory death of adult zebrafish by CML toxicity, a co-injection of apoA-I+CIGB-258 induced the highest survivability, lowest hepatic hyperinflammation, and lowest IL-6 levels (Figure 8, Figure 9, Figure 10 and Figure 11). With the lowest ROS production and apoptosis extent in the liver, the co-injection of apoA-I+CIGB-258 resulted in the most desirable lipid profile with the highest HDL-C and least hepatic damage (Figure 12 and Figure 13). These results suggest that the advantageous roles of apoA-I could be protected and enhanced by the co-presence of CIGB-258 via a putative interaction.

These synergistic interactions of apoA-I and CIGB-258 might help strengthen the antiviral activity. During infection and inflammation, a decrease in HDL-C and an increase in serum amyloid A, which can displace apoA-I from HDL, are major components of the acute-phase response. Although lipid-free apoA-I could bind with the dengue virus during attachment, apoA-I can neutralize nonstructural protein (NS)-1-induced cell activation and prevent NS-1-mediated dengue virus infection [28]. Indeed, native HDL and apoA-I displayed potent virus-killing activity against SARS-CoV-2, while glycated HDL lost its antiviral activity [29]. In addition, the paraoxonase activity was significantly impaired in the glycated HDL via the modification and loss of apoA-I, suggesting that the antioxidant activity was linked with the loss of antiviral activity. In the same context, the association of elevated apoA-I glycation and reduced HDL-associated PON activity in patients with T2DM has been reported [30]. Interestingly, the PON-1 activity was positively correlated with the increase in HDL-C and apoA-I concentrations in the healthy control group but not in T2DM patients [30,31]. These results showed that the inhibition of apoA-I glycation by CML might be a suitable pharmaceutical target to maximize the antioxidant and anti-infection activity by stabilizing the tertiary structure and functionality. Indeed, the PON-1 activities in human HDL_2_ and HDL_3_ were elevated by the co-addition of apoA-I:CIGB-258, in a dose-dependent manner, of CIGB-258 (Figure 3A), suggesting that the co-presence of CIGB-258 could enforce the antiviral activity of HDL, as reported elsewhere [32].

On the other hand, CIGB-258 was found to bind with apoA-I and transthyretin (TTR) in the serum, as identified by affinity chromatography and mass spectrometry [20,33]. Interestingly, apoA-I is synthesized in the liver and intestine, whereas TTR, a transport protein for thyroxine (T4) and retinol, is synthesized in the liver and brain [34]. Although the two proteins appear to have no relation with each other, TTR has several connections to the apoA-I metabolism: (1) serum TTR circulates in HDL through binding to apoA-Il; (2) TTR can cleave the C-terminus of apoA-I; and (3) the cleaved apoA-I by TTR impaired cholesterol efflux and promoted amyloidogenesis [35]. Therefore, the co-presence of CIGB-258 with the apoA-I and TTR in the serum might suggest putative binding for protecting apoA-I, by CIGB-258, from cleavage by a TTR attack. Indeed, the amphipathic helix domain of CIGB-258, 81.4% of the α-helix content in 22mer amino acid within the 5–26 residue, was similar to an amphipathic helix domain of apoA-I, 74.9% of the α-helix in the entire sequence. In more detail, eight helix domains consisted of 22mer amino acids in apoA-I, which are highly homologous with the α-helix of CIGB-258. Future research should be carried out to determine which helix domain of apoA-I could bind to CIGB-258 to understand the physiological meaning of the co-presence of CIGB-258 between apoA-I and TTR. Furthermore, the next in vivo studies should be conducted separately on male and female zebrafish to determine the sex-based response of apoA-I and CIGB-258.

## 4. Materials and Methods

### 4.1. Materials

Jusvinza^®^ (CIGB-258) is a lyophilized powder formulation containing the recombinant peptide derived from HSP60, consisting of 27 amino acids (Lot# 1125J1/0; 1.25 mg/vial). The peptide was obtained from the Center for Genetic Engineering and Biotechnology (CIGB) in Havana, Cuba, for exclusive research use. Unless otherwise noted, all other chemicals and reagents were of analytical grade and used as supplied.

### 4.2. Isolation of Lipoprotrins from the Blood

The density gradient ultracentrifugation technique was used to isolate lipoprotein (LDL and HDL) from human blood [36]. First, the serum was collected from the blood and subsequently processed for density gradient centrifugation in a density gradient mixture of NaCl (1.019 < *d* < 1.063) from LDL and (1.063 < *d* < 1.225) for HDL. The isolated LDL and HDL were processed for overnight dialysis using Tris-buffered saline (pH 8.0) and stored at −21 °C for further use. A detailed procedure is outlined in Appendix A.

### 4.3. Purification of Human apoA-I

Apolipoprotein A-I (apoA-I) was extracted from the HDL using a previously described method [37], utilizing fast protein liquid chromatography with an AKTA purifier system (GE Healthcare, Uppsala, Sweden). The SDS-PAGE was performed to confirm the purity of the separated apoA-I. A detailed procedure is outlined in Appendix A.

### 4.4. Effect of CIGB-258 on the Oxidation of LDL

The ability of apoA-I and the CIGB-258 mixture to prevent LDL oxidation was assessed using a thiobarbituric acid reactive substance (TBARS) assay [38] with malondialdehyde (MDA) standard and performing 0.5% agarose gel electrophoresis [39]. In brief LDL (1 mg/mL) was treated with only CuSO_4_ (10 μM) for 4 h or with the apoA-I:CIGB-258 mixture with molar ratios of 1:0, 1:0.1, 1:0.5, and 1:1. In addition the effect of different concentrations of CIGB-258 in presence of 2 mg/mL of HDL_2_ and HDL_3_ was evaluated on the CuSO_4_ (10 μM) mediated oxidation. A detailed procedure is outlined in Appendix A.

### 4.5. Measurement of Trp Fluorescence of apoA-I during Isothermal Denaturation

The WMF of tryptophan (Trp) in apoA-I in the co-presence of CIGB-258 was determined using a previously described method [16,40]. Briefly, to minimize tyrosine fluorescence interference, samples were excited at 295 nm, and emission spectra were registered from 305 to 400 nm. Isothermal denaturation experiments were performed to assess the impact of varying urea concentration (0 M to 7 M) on the secondary structures of apoA-I and CIGB-258, employing molar ratios of 1:0, 1:0.1, 1:0.5, and 1:1 in the lipid-free state. The exposure extent of Trp in apoA-I during denaturation in the presence of CIGB-258 was measured independently by fluorescence spectroscopy using the earlier described method [41].

### 4.6. Paraoxonase Assay

The HLD-associated paraoxonase (PON)-1 activity was determined by the hydrolysis of paraoxon into p-nitrophenol and diethylphosphate catalyzed by the ezyme, following the earlier described method [42]. A detailed methodology is outlined in Appendix A.

### 4.7. Anti-Glycation Activity of CIGB-258

The antiglycation effect was assessed by incubating a mixture of apoA-I:CIGB-258 at molar ratios of 1:0, 1:0.1, 1:0.5, and 1:1 [apoA-I (85 mg in 340 mL) and CIGB-258 (0.9 mg, 4.5 mg, and 8.9 mg in 60 mL)] in the presence of 200 μM CML (100 μL). After 72 h incubation, fluorescence at the excitation wavelength (370 nm) and emission wavelength (440 nm) was used to assess the degree of advanced glycation reactions, as described elsewhere [41]. The WMF measurement also evaluated the movement of Trp during glycation at excitation = 295 nm and emission = 305–400 nm [40]. A detailed method is outlined in Appendix A.

### 4.8. Protein Determination in Lipoprotein

The Lowry method, reported by Markwell et al. [42], was used to quantify the protein in the HDL and LDL. The Quick Start™ Bradford Protein Assay Kit (Bio-Rad #5000201; Hercules, CA, USA) was utilized to quantify the lipid-free apoA-I protein. Bovine serum albumin (BSA) was used as a reference.

### 4.9. Zebrafish Maintenance, Mating and Embryo Production

The zebrafish and their embryos were maintained following the standard protocol [43] and the Guide for the Care and Use of Laboratory Animals [44] approved by the Raydel Research Institute (approval code RRI-20-003, Daegu, Republic of Korea). Male and female zebrafish were mixed 2:1 for embryo production and allowed uninterrupted mating. Produced embryos were collected and processed for microinjection.

### 4.10. Microinjection of CML and apoA-I and CIGB-258 into Zebrafish Embryos

At 1 h post-fertilization (hpf), zebrafish embryos underwent individual microinjections using a pneumatic picopump (PV830; World Precision Instruments, Sarasota, FL, USA). Each microinjection contained 1.4 ng of apoA-I with co-injection, including CIGB-258 at doses 14.3 pg, 71.5 ng, and 143 pg, corresponding to apoA-I:CIGB-258 molar ratios of 1:0.1, 1:0.5, and 1:1, respectively, together with CML (500 ng). All injections were performed at the same site to reduce bias, adopting the earlier defined method [16,17]. Instantly, after injection, up to 48 h post-injection, embryos were visualized under a microscope to determine mortality and developmental changes. The used doses for the microinjection were selected based on the screening study where different amounts were injected into the embryos to assess their effect on the embryo’s survivability Appendix A. 

### 4.11. Visualizing Oxidative Stress and Apoptosis in the Embryos

The reactive oxygen species (ROS) and apoptosis in embryos from different groups were examined using dihydroethidium (DHE) and acridine orange (AO) fluorescent staining, following the previously described method [45,46]. The stained embryos were visualized under a fluorescent microscope to detect DHE and AO fluorescent intensities. Quantification of the fluorescent intensities was performed using Image J software version 1.53r (http://rsb.info.nih.gov/ij/, accessed on 15 December 2023). A detailed methodology is provided in Appendix A.

### 4.12. Anti-Inflammatory Activity of apoA-I and CIGB-258 in Zebrafish

To induce acute inflammation in zebrafish, an intraperitoneal injection of carboxymethyllysine (CML) was administered, with a dose of 250 μg/10 μL of PBS (equating to a final concentration of approximately 3 mM based on an average zebrafish body weight of 300 mg [16]. The zebrafish were randomly assigned to 5 groups (n = 40 for each group). Group I received an injection of 10 μL PBS, serving as the control. Group II was injected with 250 μg of CML/10 μL PBS. Groups III, IV, and V were given 250 μg CML/10 μL of PBS combined with either apoA-I (8.5 μg), CIGB-258 (0.9 μg), or a combination of apoA-I (8.5 μg) + CIGB-258 (0.9 μg), respectively. The used doses for the microinjection were selected based on the screening study where different amounts were injected into the embryos to assess their effect on the embryo’s survivability Appendix A.

The swimming activity across all the groups was assessed at 30 min and 60 min post-treatment by previously established parameters [47]. Three hours post-treatment, the zebrafish across all the groups were sacrificed, and blood was promptly collected. The blood was then processed to measure aspartate transaminase (AST) and alanine transaminase (ALT) levels, using commercial kits from Asan Pharmaceutical (Hwasung, Republic of Korea), following the manufacturer’s instructions.

### 4.13. Liver Histology and Immunohistochemistry

Liver tissue was surgically extracted from each experimental group and fixed in 10% formalin for 24 h. Following dehydration in alcohol, the tissue was embedded in paraffin, and 5 μm-thick sections were prepared using a microtome. These were then processed to assess morphological changes through H&E staining and IL-6 production via immunohistochemical staining, as detailed in Appendix A.

### 4.14. Statistical Evaluation

The data are expressed as the mean ± SEM for three independent experiments. The group comparisons and statistical analyses were conducted using a one-way analysis of variance (ANOVA) with SPSS software (version 29.0, SPSS, Inc., Chicago, IL, USA), followed by a Tukey’s multiple range test for post hoc analyses.

## 5. Conclusions

The co-presence of CIGB-258 improved the structural stability of apoA-I to display enhanced beneficial functionality, such as antioxidant and anti-glycation activities. CIGB-258 could interact well with apoA-I to stabilize its amphipathic helices and maintain the tertiary structure of apoA-I. Combining apoA-I and CIGB-258 enhanced the in vitro antioxidant ability to prevent LDL oxidation and enforce HDL-associated paraoxonase activity. The co-presence of CIGB-258 protected apoA-I from CML-mediated glycation by stabilizing the tertiary structure. The combination of apoA-I and CIGB-258 synergistically increased the in vivo anti-inflammatory activity against CML-induced toxicity in zebrafish adults and embryos. Overall, the co-presence of apoA-I and CIGB-258 exhibited remarkable synergistic impacts, augmenting the structural and functional association of HDL in a dose-dependent manner. The study’s results serve as a valuable therapeutic strategy for managing inflammatory-related conditions.

## Figures and Tables

**Figure 1 ijms-25-05560-f001:**
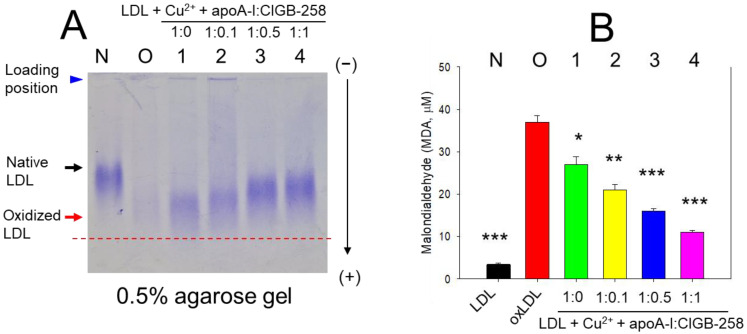
Effect of apolipoprotein (apoA-I) and CIGB-258 in protecting cupric ion-mediated low-density protein (LDL) oxidation. (**A**) Electrophoretic mobility of LDL (10 μg of protein) was analyzed in the presence of Cu^2+^ (final 1 μM), along with apoA-I and CIGB-258. The electrophoresis was conducted at 50 V for 1 h using pH 8.0-Tris-EDTA buffer. The gel was stained with 1.25% Coomassie brilliant blue. Lane N and Lane O, representing native LDL and oxidized LDL (LDL + Cu^2+^), respectively; lane 1, LDL + Cu^2+^ + apoA-I:CIGB-258 (1:0); lane 2, LDL + Cu^2+^ + apoA-I:CIGB-258 (1:0.1); lane 3, LDL + Cu^2+^ + apoA-I:CIGB-258 (1:0.5); lane 4, LDL + Cu^2+^ + apoA-I:CIGB-258 (1:1). (**B**) Thiobarbituric acid reactive substance (TBARS) assessment in LDL exposed to cupric ions and later subjected to a mixture of apoA-I and CIGB-258 was quantified by TBARS assay employing malondialdehyde as reference. Statistical significance is denoted by *, ** and *** at *p* < 0.05, *p* < 0.01 and *p* < 0.001, compared to the LDL + Cu^2+^ (ox-LDL). Red dotted line indicates position of oxLDL electromobility.

**Figure 2 ijms-25-05560-f002:**
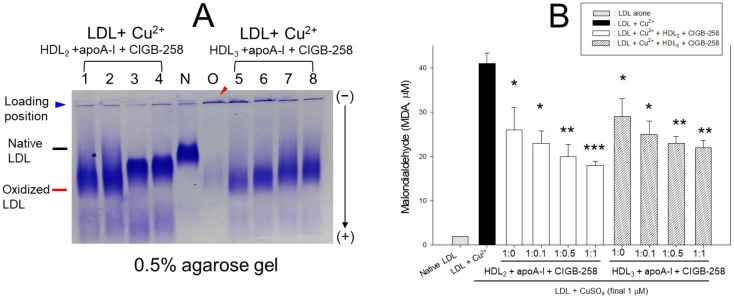
High-density lipoprotein (HDL) antioxidant ability against low-density liprotein (LDL) oxidation was enhanced by adding CIGB-258 in a concentration-responsive manner. (**A**) Electromobility of cupric ion-mediated oxidized LDL in the presence of HDL and CIGB-258. Lane N, native LDL; lane O, oxidized LDL (LDL + Cu^2+^, final 1 μM); lane 1, LDL + Cu^2+^ + HDL_2_ + apoA-I:CIGB-258 (1:0); lane 2, LDL + Cu^2+^ + HDL_2_ + apoA-I:CIGB-258 (1:0.1); lane 3, LDL + Cu^2+^ + HDL_2_ + apoA-I:CIGB-258 (1:0.5); lane 4, LDL + Cu^2+^ + HDL_2_ + apoA-I:CIGB-258 (1:1); lane 5, LDL + Cu^2+^ + HDL_3_ + apoA-I:CIGB-258 (1:0); lane 6, LDL + Cu^2+^ + HDL_3_ + apoA-I:CIGB-258 (1:0.1); lane 7, LDL + Cu^2+^ + HDL_3_ + apoA-I:CIGB-258 (1:0.5); lane 8, LDL + Cu^2+^ + HDL_3_ + apoA-I:CIGB-258 (1:1). Red arrowhead indicates aggregated oxLDL band in loading position. (**B**) Assessement of oxidized species using a thiobarbituric acid reactive substance (TBARS) assay employing malondialdehyde as reference. Statistical significance is denoted by *, ** and *** at *p* < 0.05, *p* < 0.01 and *p* < 0.001, compared to the LDL + Cu^2+^ (ox-LDL).

**Figure 3 ijms-25-05560-f003:**
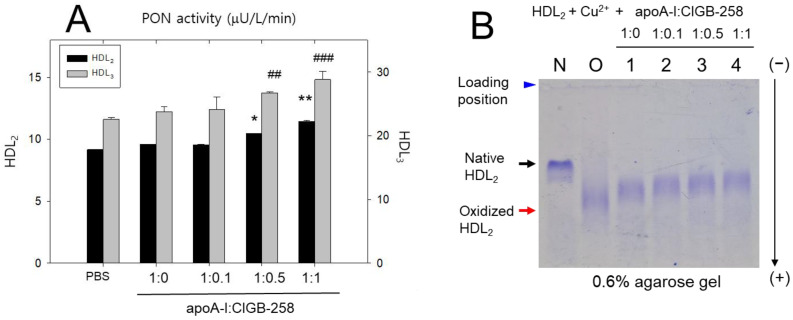
Improvement in the antioxidant ability of high-density lipoprotein (HDL) by co-addition of an apoA-I:CIGB-258 mixture. (**A**) Paraoxonase (PON) activity of HDL_2_ and HDL_3_ in the presence of an apoA-I:CIGB-258 mixture. HDL_2_ and HDL_3_, each diluted to a concentration of 1 mg/mL, were added to 0.18 mL of a paraoxon-ethyl (Sigma catalog number D-9286) solution (pH 8.5) comprising Tris–HCl (90 mM), NaCl (3.6 mM), and 2 mM CaCl_2_ (2 mM) for 60 min. (**B**) Electromobility of cupric ion-mediated oxidized HDL in the presence of apoA-I and CIGB-258. Lane N, native HDL_2_; lane O, oxidized HDL_2_; lane 1, HDL_2_ + Cu^2+^ + apoA-I:CIGB-258 (1:0); lane 2, HDL_2_ + Cu^2+^ + apoA-I:CIGB-258 (1:0.1); lane 3, HDL_2_ + Cu^2+^ + apoA-I:CIGB-258 (1:0.5); lane 4, HDL_2_ + Cu^2+^ + apoA-I:CIGB-258 (1:1). Statistical significance is denoted by * and ** at *p* < 0.05, and *p* < 0.01, for PON activity observed in HDL_3_, while ^##^ and ^###^ denote the statistical significance at *p* < 0.01, and *p* < 0.001, for PON activity observed in HDL_2_ compared to the PBS group.

**Figure 4 ijms-25-05560-f004:**
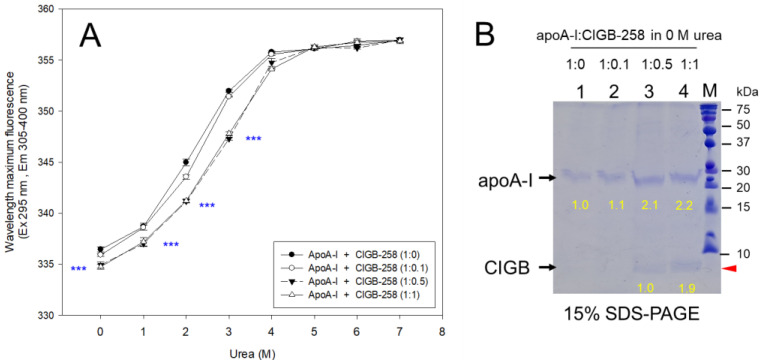
Isothermal denaturation of apoA-I and CIGB-258 in the lipid-free state by the urea treatment for 16 h incubation. (**A**) Change in the wavelength of maximum fluorescence (WMF) in apoA-I with different molar ratios of apoA-I:CIGB-258 during the urea treatment. The increase in urea concentration was used to assess the change in Trp exposure (excitation at 295 nm, emission spectra range 305–400 nm), presented as WMF. (**B**) Electrophoresis pattern of the apoA-I and CIGB-258 mixture at 0 M urea. Electrophoretic profiles of the apoA-I:CIGB-258 mixture were visualized by staining the protein bands with 0.125% Coomassie brilliant blue. The yellow numbers indicate the band intensity of each band. The red arrowhead indicates the band of CIGB-258 at the bottom of the gel. Lanes 1, 2, 3 and 4 represent apoA-I + CIGB-258 at the molar ratios of 1:0, 1:0.1, 1:0.5 and 1:1, respectively. Statistical significance *** denotes *p* < 0.001 between 1:0 and 1:1 of apoA-I:CIGB-258.

**Figure 5 ijms-25-05560-f005:**
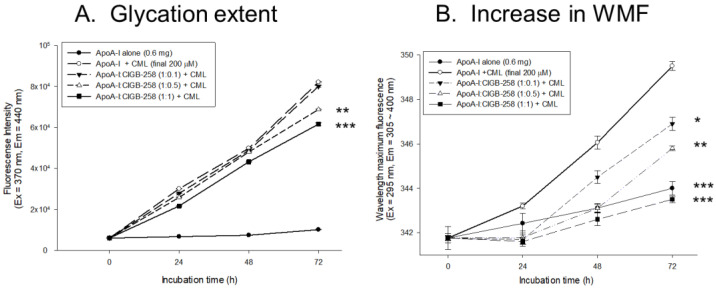
Effect of CIGB-258 in countering CML-mediated glycation of apolipoprotein (apoA-I). (**A**) The fluorescent intensity was assessed under excitation wavelength (Ex) and emission wavelength (Em) of 370 nm and 440 nm, respectively. The measurement was conducted over a 72 h incubation period in the presence of 200 μM CML. (**B**) The degree of tryptophan (Trp) exposure in apoA-I was evaluated over the 72 h glycation process. Changes in the wavelength of maximum fluorescence (WMF) in apoA-I were observed across the molar ratios of apoA-I:CIGB-258 during CML-mediated glycation. As the extent of glycation and incubation time increased, alterations in Trp exposure were compared using WMF (Ex = 295 nm, Em range = 305–400 nm). Statistical significance is denoted by *, ** and *** at *p* < 0.05, *p* < 0.01 and *p* < 0.001, compared to the apoA-I + CML group.

**Figure 6 ijms-25-05560-f006:**
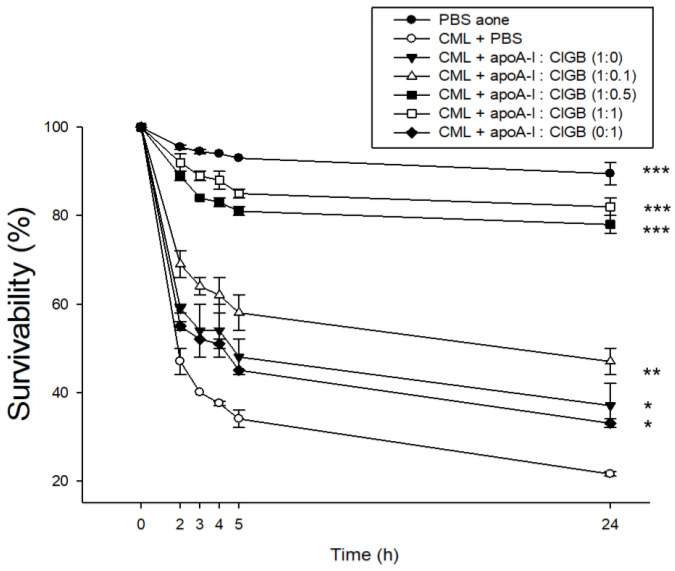
Effect of different combinations of apolipoprotein A-I (apoA-I) and CIGB-258 on the survivability of carboxymethyllysine (CML)-treated zebrafish embryos during 24 h post-treatment. PBS and CML+PBS group received 10 nL microinjection of PBS and 500 ng CML in PBS, respectively; apoA-I:CIGB-258 (1:0) and apoA-I:CIGB-258 (0:1) group injected with 1.4 ng/10 nL apoA-I, and 143 pg/10 nL CIGB-258, respectively; apoA-I:CIGB-258 (1:0.1) or (1:0.5) or (1:1) injected with 10 nL of 1.4 ng apoA-I containing 14 pg or 70 pg or 143 pg for CIGB-258. Statistical significance denoted by *, ** and *** at *p* < 0.05, *p* < 0.01 and *p* < 0.001, compared to the CML+PBS group.

**Figure 7 ijms-25-05560-f007:**
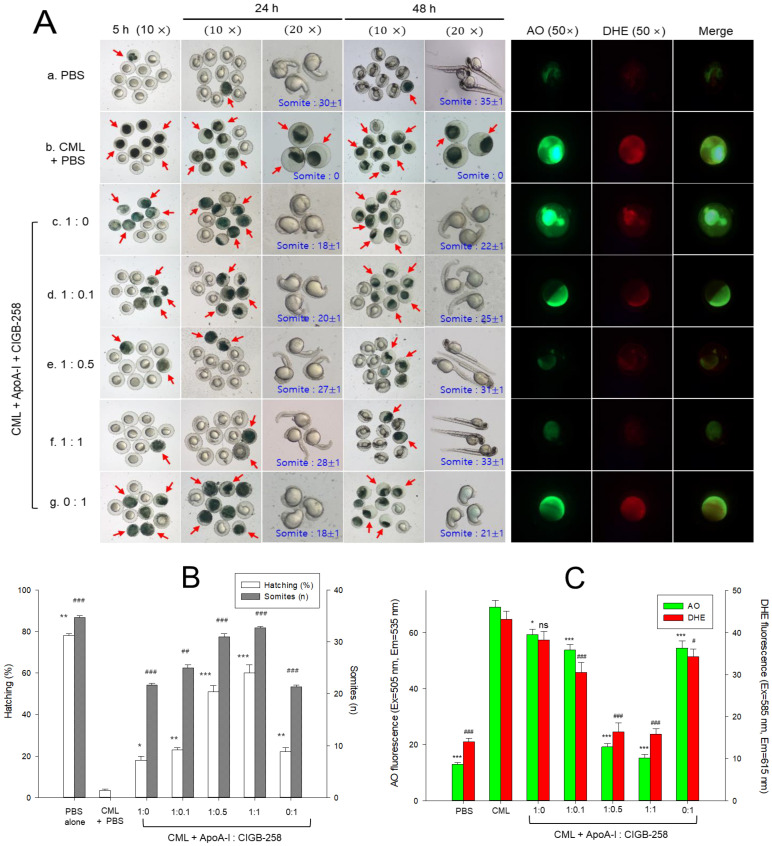
Effect of different combinations of apolipoprotein A-I (apoA-I) and CIGB-258 on the developmental changes, reactive oxygen species generation (ROS) and apoptosis in carboxymethyl lysine (CML) treated zebrafish embryos. (**A**) Pictorial view of the developmental changes during 5–48 h post-treatment (the red arrow indicates the dead embryos). ROS and apoptosis levels were determined by dihydroethidium (DHE) and acridine orange (AO) fluorescent staining (images were captured at 5 h post-treatment). (**B**) Hatching (%) and somites (n) depicting quantification of developmental changes at 24 h post-treatment. (**C**) Image J-based quantification of AO and DHE fluorescent intensities. PBS and CML+PBS group received 10 nL microinjection of PBS and 500 ng CML in PBS, respectively; apoA-I:CIGB-258 (1:0) and apoA-I:CIGB-258 (0:1) group injected with 1.4 ng/10 nL apoA-I, and 143 pg/10 nL CIGB-25, respectively; apoA-I:CIGB-258 (1:0.1) or (1:0.5) or (1:1) injected with 10 nL of 1.4 ng apoA-I containing 14 pg or 70 pg or 143 pg for CIGB-258. Statistical significance denoted by *, ** and *** at *p* < 0.05, *p* < 0.01 and *p* < 0.001 for AO fluorescent and hatching (%); ^#^, ^##^, and ^###^ at *p* < 0.05, *p* < 0.01 and *p* < 0.001 for DHE fluorescent and somite counts were compared to the CML+PBS group, respectively; ns represent nonsignificant difference between the groups.

**Figure 8 ijms-25-05560-f008:**
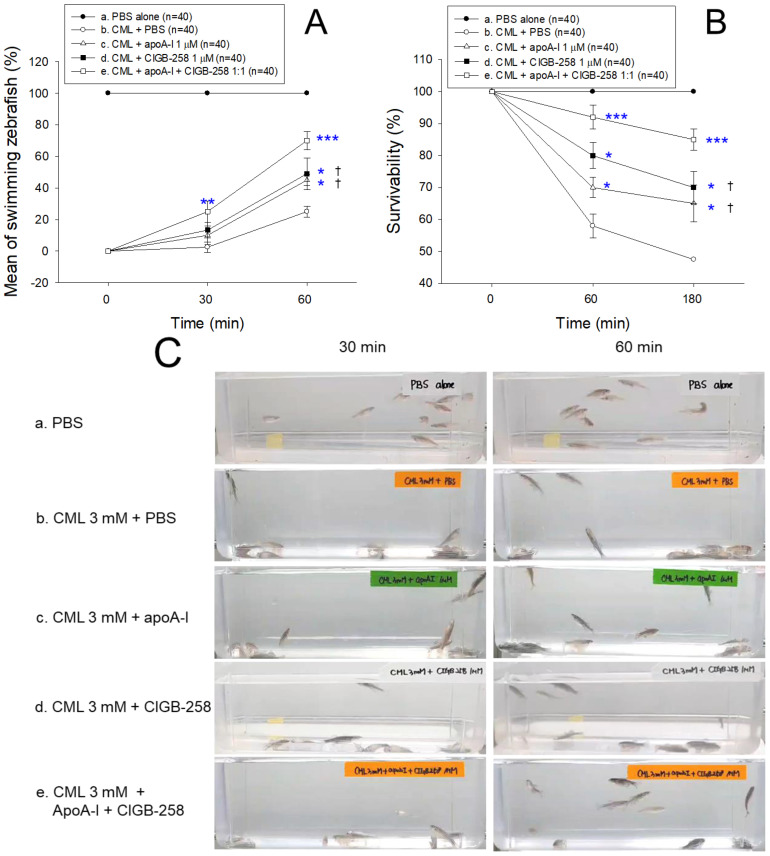
Effect of different combinations of apolipoprotein A-I (apoA-I) and CIGB-258 on the swimming ability and survivability of zebrafish injected with carboxymethyllysine (CML). (**A**) Mean swimming activity at 30 min and 60 min post-treatment. (**B**) Survivability of zebrafish at 60 min and 180 min post-treatment. (**C**) Pictorial view of the zebrafish at 30 min and 60 min post-treatment. PBS and CML+PBS group received 10 μL microinjection of PBS and 250 μg CML (3 mM) in PBS, respectively; apoA-I, CIGB-258 and apoA-I+CIGB-258 groups injected with 10 μL of 250 μg CML with 8.5 μg apoA-I (1 μM) or 0.9 μg CIGB-258 (1 μM), or 8.5 μg apoA-I (1 μM) + 0.9 μg CIGB-258 (1 μM), respectively. Statistical significance denoted by *, ** and *** at *p* < 0.05, *p* < 0.01 and *p* < 0.001, compared to the CML+PBS group, respectively, while ^†^ at *p* < 0.05 compared to the apoA-I+CIGB-258 group.

**Figure 9 ijms-25-05560-f009:**
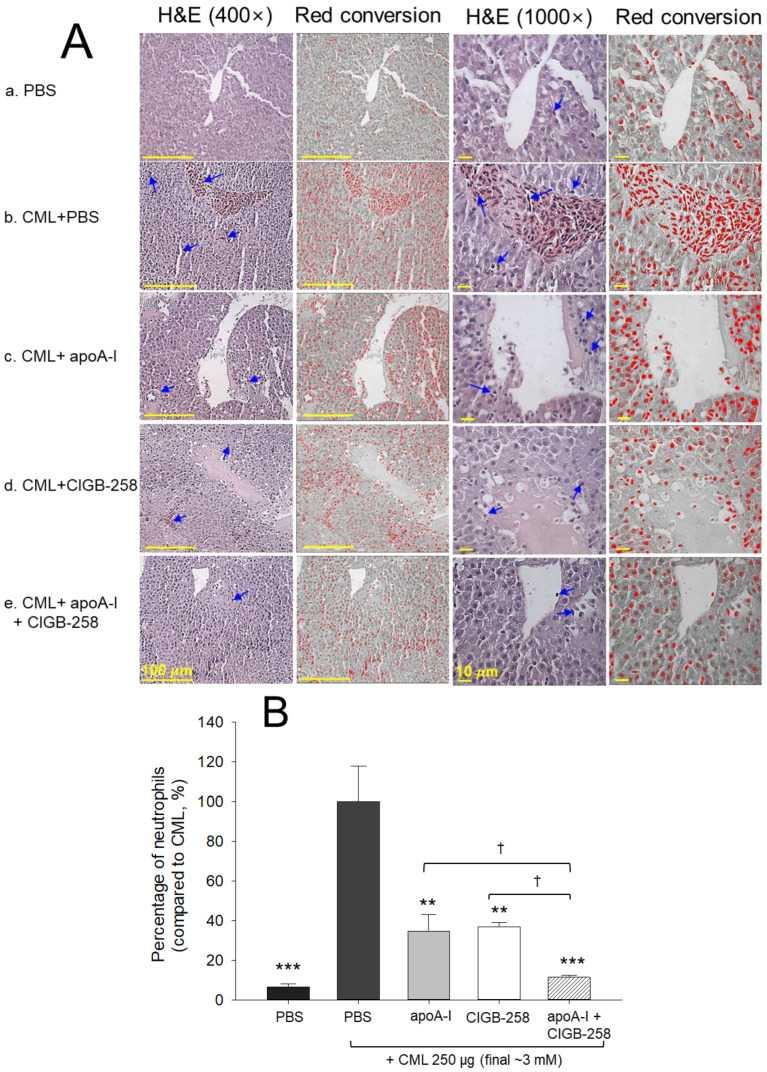
Liver histological analysis of zebrafish injected with carboxymethyllysine (CML) and subsequently treated with different combinations of apolipoprotein A-I (apoA-I) and CIGB-258. (**A**) Hematoxylin and Eosin (H&E) staining visualized at 100× [scale bar = 100 μm] and 1000× [scale bar = 10 μm] magnification. The blue arrow indicates neutrophils [scale bar = 100 μm]. (**B**) Percentage of neutrophils counted in H&E-stained area. A semiquantitative assessment of neutrophils (stained dark-violet color), was assessed by microscopic examination of designated area (1.23 mm^2^) across three distinct sections and five different areas of each group. PBS and CML+PBS group received 10 μL microinjection of PBS and 250 μg CML (3 mM) in PBS, respectively; apoA-I, CIGB-258 and apoA-I+CIGB-258 groups injected with 10 μL of 250 μg CML with 8.5 μg apoA-I (1 μM) or 0.9 μg CIGB-258 (1 μM), or 8.5 μg apoA-I (1 μM) + 0.9 μg CIGB-258 (1 μM), respectively. Statistical significance denoted by ** and *** at *p* < 0.01 and *p* < 0.001. compared to the CML+PBS group, while ^†^ at *p* < 0.05 compared to the apoA-I+CIGB-258 group.

**Figure 10 ijms-25-05560-f010:**
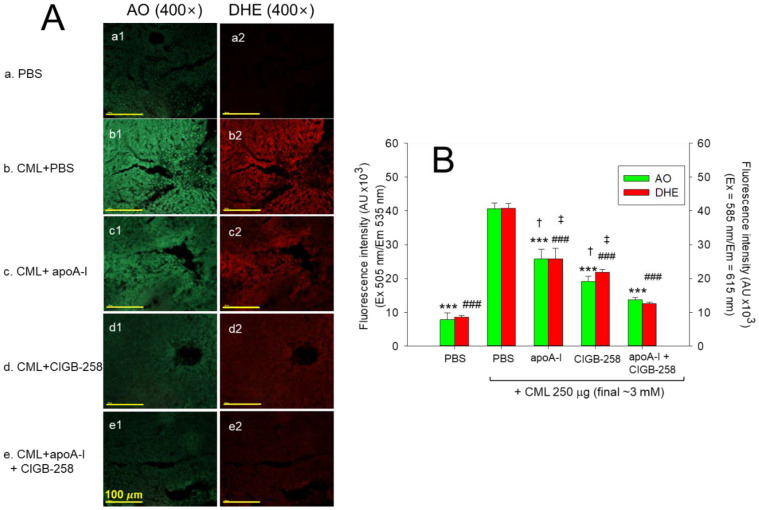
Reactive oxygen species (ROS) and extent of apoptosis in the liver section of zebrafish injected with carboxymethyllysine (CML) and subsequently treated with different combinations of apolipoprotein A-I (apoA-I) and CIGB-258. (**A**) Fluorescent images of dihydroethidium (DHE)- and acridine orange (AO)-stained area. (**B**) Quantification of DHE and AO fluorescent intensities employing Image J software. PBS and CML+PBS group received 10 μL microinjection of PBS and 250 μg CML (3 mM) in PBS, respectively; apoA-I, CIGB-258 and apoA-I+ CIGB-258 groups injected with 10 μL of 250 μg CML with 8.5 μg apoA-I (1 μM) or 0.9 μg CIGB-258 (1 μM), or 8.5 μg apoA-I (1 μM) + 0.9 μg CIGB-258 (1 μM), respectively. Statistical significance denoted by *** and ^###^ at *p* < 0.001 for AO and DHE fluorescent intensities, respectively, compared to the CML+PBS group, while ^†^ and ^‡^ at *p* < 0.05 compared to the apoA-I+CIGB-258 group.

**Figure 11 ijms-25-05560-f011:**
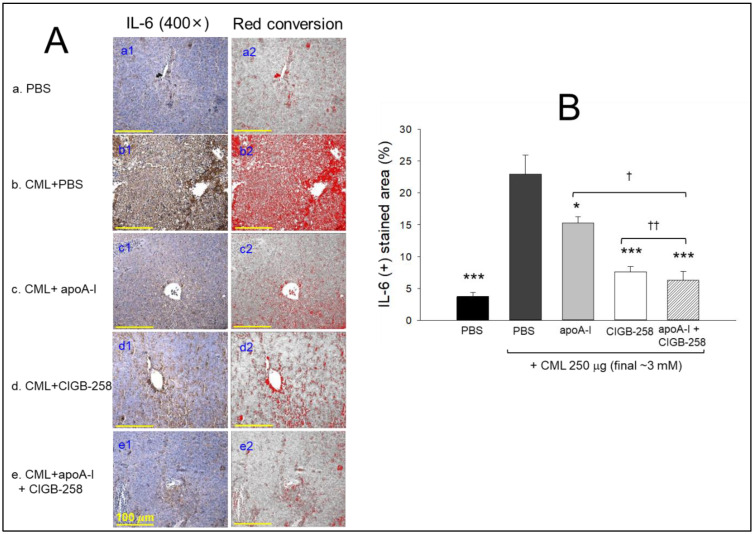
Interleukin (IL)-6 production in the liver section of zebrafish injected with carboxymethyllysine (CML) and subsequently treated with different combinations of apolipoprotein A-I (apoA-I) and CIGB-258. (**A**) Images of immunohistochemistry (**a1**–**e1**). Red conversion images (**a2**–**e2**) are IHC-stained areas (brown color) interchanged with red color [at a threshold value of (20–100)] using Image J software to enhance visualization. (**B**) Quantification of IL-6-stained area employing Image J software. PBS and CML+PBS group received 10 μL microinjection of PBS and 250 μg CML (3 mM) in PBS, respectively; apoA-I, CIGB-258 and apoA-I+CIGB-258 groups injected with 10 μL of 250 μg CML with 8.5 μg apoA-I (1 μM) or 0.9 μg CIGB-258 (1 μM), or 8.5 μg apoA-I (1 μM) + 0.9 μg CIGB-258 (1 μM), respectively. Statistical significance denoted by * and *** at *p* < 0.05 and *p* < 0.001 compared to the CML+PBS group, while ^†^ and ^††^ at *p* < 0.05 and *p* < 0.01, compared to the apoA-I+CIGB-258 group.

**Figure 12 ijms-25-05560-f012:**
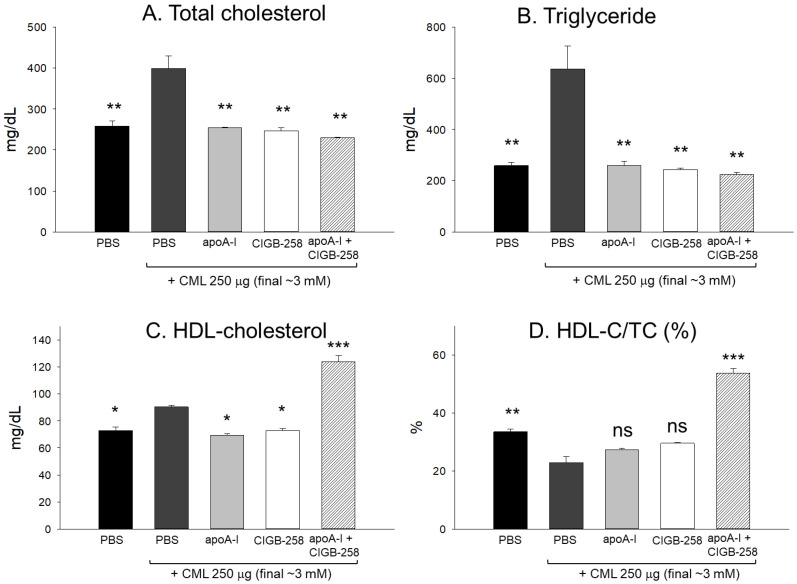
Comparison of plasma lipid profiles. (**A**) total cholesterol (TC), (**B**) triglyceride (TG), (**C**) high-density lipoprotein cholesterol (HDL-C), and (**D**) HDL-C/TC (%) in the serum of zebrafish injected with carboxymethyllysine (CML) alone or in conjunction with apoA-I, CIGB-258 and apoA-I+CIGB-258. Statistical significance is indicated by *, ** and *** at *p* < 0.05, *p* < 0.01 and *p* < 0.001, compared to the CML+PBS group; ns is a non-significant difference between the groups.

**Figure 13 ijms-25-05560-f013:**
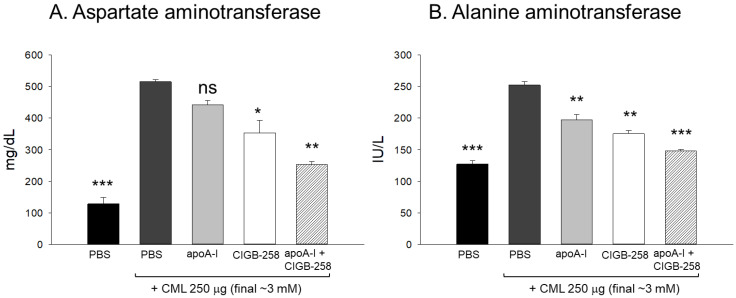
Measurement of the zebrafish plasma hepatic enzyme levels (**A**) aspartate transaminase (AST) and (**B**) alanine transaminase (ALT) detected at 180 min post-injection of CML alone or in combination with apoA-I, CIGB-258 and apoA-I+CIGB-258. Statistical significance is denoted by *,** and *** at *p* < 0.05, *p* < 0.01 and *p* < 0.001, compared to the CML+PBS group; ns is a non-significant difference between the groups. AST refers to aspartate aminotransferase, ALT to alanine aminotransferase, and CML to carboxymethyl lysine.

**Table 1 ijms-25-05560-t001:** Alternation in the median wavelength of maximum fluorescence (WMF) during urea-induced isothermal denaturation.

Protein + Peptide	Molar Ratio apoA-I:CIGB-258 ^1^	Median WMF (nm)	D_1/2_ ^2^ Urea, (M)	*r*	*p*
apoA-I alone	1:0	346.2	2.33	0.990	0.001
apoA-I + CIGB-258	1:0.1	345.9	2.35	0.988	0.001
apoA-I + CIGB-258	1:0.5	345.7	2.57	0.977	0.001
apoA-I + CIGB-258	1:1	345.5	2.56	0.980	0.001
CIGB-258 alone	0:1	329.1	4.77	0.976	0.001

^1^ WMF of apoA-I at the median urea concentration of 3.5 M. ^2^ D_1/2_ value represents the urea concentration required to achieve 50% denaturation of apoA-I, determined by regression analysis as the mid-point of denaturation, apolipoprotein A-I (ApoA-I, MW = 28,078), and CIGB-258 (Jusvinza, MW = 2987).

## Data Availability

The data used to support the findings of this study are available from the corresponding author upon reasonable request.

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
