# Peer review of "Synergistic Anti-Inflammatory Activity of Lipid-Free Apolipoprotein (apo) A-I and CIGB-258 in Acute-Phase Zebrafish via Stabilization of the apoA-I Structure to Enhance Anti-Glycation and Antioxidant Activities"

_ijms, 2024, doi:10.3390/ijms25105560_

Round 1

Reviewer 1 Report

Comments and Suggestions for Authors

 Minor bugs:

I propose to include a list of terms - abbreviations of the names used in the article. The text must be carefully checked for spelling: e.g. 173, 174 lines, a fragment of the text is in a different font, sometimes there are double spaces.

Figure 6 - maybe it can be divided into separate drawings. Some drawing captions are very long - maybe they can be shortened. Can the authors include anything else regarding the research results in their conclusions?

The main problem with this article is that it contains too many borrowings from other articles. This is visible in the methodology, but also in the captions to the figures and in the text. 

Author Response

Thank you for your valuable comments and suggestions. 

Please find attached doc as point-to-point response.

Reviewer 2 Report

Comments and Suggestions for Authors

Cho et al studied the synergistic anti-inflammatory activity of lipid-free apolipoprotein (apo) A-I and CIGB-258. The results are of interest. However, there are some concerns need t be addressed.

1.       Line 246: It is subheading as “2.5 Synergistic Anti-Glycation Activity of apoA-I and CIGB-258”. However, the experiment is not about the synergy between apoA-I and CIGB-258. Rather it studied anti-glycation efficacy of CIGB-258 against glycation of apoA-I. So, the subheading should be modified.

2.       Figure 6 study the synergistic embryo protection ability of apoA-I and CIGB-258 against carboxymethyl-lysine (CML)-induced acute embryo death. However, in the experiment, the group of CIGB-258 alone is missing. Therefore, the synergistic effect cannot be confidently concluded. For example, the results could be due to a stronger effect of CIGB-258 than apoA-I, rather than synergy. Therefore, CIGB-258 alone should be included in the study.

3.       Figure 7A and 7B, only compared to control. The authors need to compare the CIGB-258 + apoA-I group with the CIGB-258 alone group (as well the apoA-I along group) to conclude synergy. Currently these comparisons are missing.

4.       Figure 8: What does the blue area stand for? The entire section of “2.8 histologic Analysis of Hepatic Tissue” is flawed. H & E staining is a neutrophil staining. In addition, the area calculation is also flawed, as the empty space, which may be blood vessel or other structures, affects the results.

5.       Figure 9 needs to compare the last column (combined) with the second and third last group.

6.       Figure 10 needs to compare the last column (combined) with the second and third last group.

Author Response

(The authors gave the same response as above.)

Reviewer 3 Report

Comments and Suggestions for Authors The manuscript presented from Cho et al., entitled "Synergistic anti-inflammatory activity of lipid-free apolipoprotein (apo) A-I and CIGB-258 in acute phase zebrafish via stabilization of apoA-I structure to enhance anti-glycation and antioxidant activities" is interesting and original, however there are different critical point to be addressed: 1) The authors used embryos and adult zebrafish in the present study, however the results presented in fugure 6 must be improved, the images are really small. The authors should furnish more image of high quality. 2) The authors to detect ROS used a general probe, they should use a specific kit to detect ROS such as Cell-ROX and MitoSOX to define the exact source of the oxidative stress in the embryo and in adult tissue.  it is very strange that from the figure shown in the manuscript only the yolk in red is visible. (Figure 6B) 3) The authors should explain how they choose the concentration to be injected in embryo, it is not clear, have the authors performed a toxicological curve? 4) Why the authors choose only to perform hystology and immuno for hepatic tissue? 5) The authors mention that their model develop neurotoxicity, but they did not showed brain section (electron microscopy), or performed an analysis of neuronal cells. They must improve this point. 6) Have the authors performed a separation of adults between male and female to understand if there are different effect sex-dependent? 7) Legend should be better explained, which is the number of animal used in figure 6-7-8-9?          

Author Response

(The authors gave the same response as above.)

Round 2

Reviewer 1 Report

Comments and Suggestions for Authors

I accept the article for printing in the form presented. The authors made the suggested corrections.

Reviewer 2 Report

Comments and Suggestions for Authors

The authors have addressed my concerns

Reviewer 3 Report

Comments and Suggestions for Authors

The authors improved the manuscript.